



# Learning predictable and informative dynamical drivers of extreme precipitation using variational autoencoders

Fiona R. Spuler[1, 2], Marlene Kretschmer[1, 3], Magdalena Alonso Balmaseda[4], Yevgeniya Kovalchuk[5], and Theodore G. Shepherd[1]

[1]Department of Meteorology, University of Reading, Reading, UK
[2]The Alan Turing Institute, London, UK
[3]Leipzig Institute for Meteorology, Leipzig University, Leipzig, Germany
[4]European Centre for Medium-Range Weather Forecasts, Reading, UK
[5]Centre for Advanced Research Computing, University College London, London, UK

**Correspondence:** Fiona R. Spuler (f.r.spuler@pgr.reading.ac.uk)

**Abstract.**

Large-scale atmospheric dynamics modulate the occurrence of extreme precipitation events and provide sources of predictability of these events on timescales ranging from days to decades. In the midlatitudes, these dynamical drivers are frequently represented as discrete, persistent and recurrent circulation regimes. However, available methods identify circulation regimes which are either predictable but not necessarily informative of the relevant local-scale impact studied, or targeted to a local-scale impact but no longer as predictable. In this paper, we introduce a generative machine learning method based on variational autoencoders for identifying probabilistic circulation regimes targeted to spatial patterns of precipitation. The method, CMM-VAE, combines targeted dimensionality reduction and probabilistic clustering in a coherent statistical model and extends a previous architecture published by the authors to allow for categorical target variables. We investigate the trade-off between regime informativeness of local precipitation extremes and predictability of the regimes at subseasonal lead times. In an application to study drivers of extreme precipitation over Morocco, we find that the targeted CMM-VAE regimes are more informative of the impact variable of interest, compared to two well-established linear approaches, while maintaining the predictability of conventional non-targeted circulation regimes in subseasonal hindcasts, hence resolving the trade-off identified in previous studies. Furthermore, the targeted regimes and their predictability are physically interpretable in terms of known subseasonal teleconnections relevant to the region, the Madden-Julian Oscillation and variability of the stratospheric polar vortex. The proposed method therefore allows to identify predictable, interpretable and locally relevant representations of regional dynamical drivers given a target variable of interest. These results highlight the potential of the method for a variety of applications, ranging from subseasonal forecasting to attribution and statistical downscaling.

## 1 Introduction

Extreme events such as heatwaves and extreme precipitation cause devastating impacts on lives and livelihoods around the world every year. Improving the forecasts of these extremes at timescales ranging from days to decades, in particular in the



context of a changing climate, can support societal resilience through measures such as improved early-warning systems, forecast-based financing, and robust climate change adaptation (Coughlan de Perez et al., 2019; Lemos et al., 2012).

The occurrence and predictability of extreme events is often modulated by regional dynamical drivers such as the North
Atlantic Oscillation over north-western Europe or the Caribbean Low-Level Jet over Central America and the Caribbean (García-Martínez and Bollasina, 2020; Scaife et al., 2008). These regional dynamical drivers are frequently predictable at extended lead times themselves, and can furthermore be modulated by teleconnections from low-frequency modes of variability in the climate system, such as the El-Niño Southern Oscillation, which act as sources of predictability at longer timescales (Ferranti et al., 2018; Le et al., 2023; Mariotti et al., 2020; Saggioro et al., 2024).

Regional dynamical drivers and associated teleconnections have therefore been used to improve extreme event predictions on a range of timescales. At medium-range lead times up to 15 days ahead, forecasts conditional on regional dynamical drivers have demonstrated improved skill (Allen et al., 2021; Mastrantonas et al., 2022; Rouges et al., 2024). At subseasonal-to-seasonal (S2S) lead times, reduced representations of dynamical drivers and empirical models of teleconnections to other modes of variability have been leveraged to improve forecast skill (Bach et al., 2024; de Fondeville et al., 2023), as well as
identify and explain so-called windows of opportunity of higher-than-average forecast skill (Dunstone et al., 2023; Mariotti et al., 2020). On climate timescales, large-scale drivers have been used to gain a physical understanding of climate model uncertainty as well as conditional predictability by building plausible storylines of future change (Harvey et al., 2023; Mindlin et al., 2023; Shepherd et al., 2018).

A well-established approach to representing regional dynamical drivers, such as jet variability in the midlatitudes, is the
identification of recurrent and persistent patterns of atmospheric circulation, so-called weather regimes (Ghil and Robertson, 2002; Hannachi et al., 2017). These regimes are commonly identified using a combination of linear dimensionality reduction and non-probabilistic clustering (see e.g. Michelangeli et al. (1995)). Over several regions, this approach has been shown to identify weather regimes that are persistent and predictable by dynamical forecast models (Dorrington et al., 2022; Falkena et al., 2022; Rouges et al., 2024; Straus, 2022) as well as useful for understanding teleconnections from, for example, the
Madden-Julian Oscillation (MJO) or Stratospheric Polar Vortex (SPV) (Cassou, 2008; Domeisen et al., 2020).

While these standard weather regimes are designed to capture the main features of the circulation over a given region such as the North Atlantic, they do not necessarily disentangle the dynamical patterns that modulate extreme impacts in a specific country or area of interest (Bloomfield et al., 2020; Wiel et al., 2019). However, the ability of regional dynamical drivers to improve forecasts across different timescales depends on both their informativeness of the local extreme impact
of interest, as well as their own predictability and ability to represent the atmospheric phase space. Therefore, conventional circulation regimes, despite their predictability, do not necessarily represent suitable regional dynamical drivers for any given target variable as they lack informativeness of the local impact. Different methods have been proposed to identify regimes targeted to a local impact, such as clustering the impact variable directly (Bloomfield et al., 2020; Ullmann et al., 2014) or filtering for extreme impact days prior to clustering (Dorrington et al., 2024). However, while the resulting regimes are more
informative of the impact studied, they were shown to compromise regime predictable (Bloomfield et al., 2021).





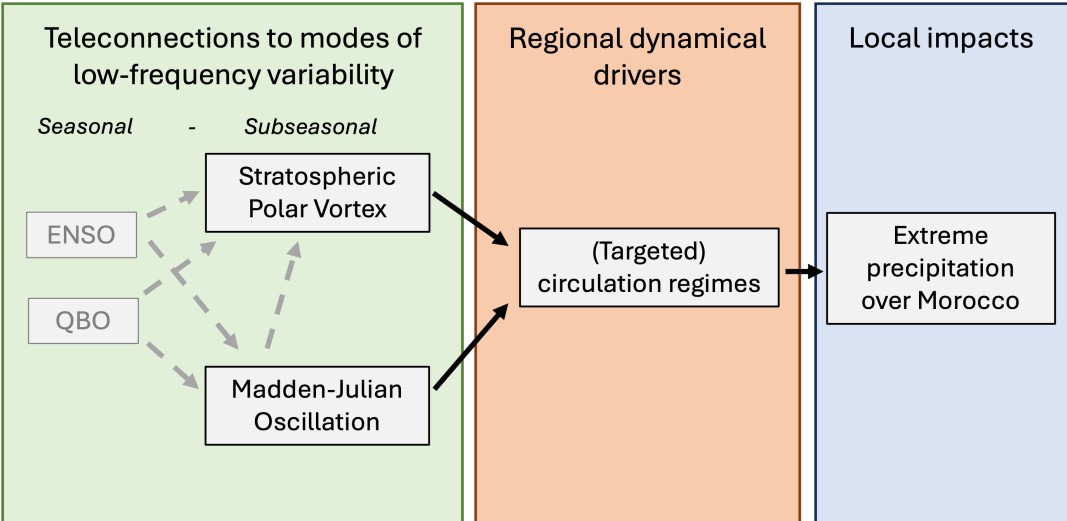

**Figure 1.** Graphical illustration of teleconnections (green box) from the SPV and MJO at subseasonal timescales, their mediation by the targeted circulation regimes (orange box) and associated impact on extreme precipitation over Morocco (blue box).

In this study, we introduce a probabilistic machine learning method for identifying targeted regimes and investigate the ability of this method to balance the trade-off between regime informativeness and predictability. The method, termed Categorical Mixture Model Variational Autoencoder (CMM-VAE), is based on a variational autoencoder architecture and combines targeted dimensionality reduction and probabilistic clustering in a single coherent statistical model that is fit using Bayesian
variational inference. The method builds on a previous method which was introduced in Spuler et al. (2024a) and will be discussed in more detail in section 2.2.3, which was found to identify regimes that are more informative of the chosen target variable while still being persistent and representative of the entire atmospheric phase space over the region. These promising results motivate the further investigation of the ability of the approach to balance the trade-off between regime predictability and informativeness found in previous targeted clustering approaches.

We apply the method to study circulation regimes targeted to precipitation over Morocco, as well as their predictability in subseasonal hindcasts and associated teleconnections (see Figure 1 for an overview). With most of the rainfall occurring in extended winter, the country is vulnerable to both extreme rainfall, which leads to flooding and is the focus of this study, as well as drought, which impacts agricultural livelihoods and overall macroeconomic stability (Loudyi et al., 2022). Previous studies have shown that extreme precipitation events over Morocco are associated with dynamically driven moisture flux from the
Atlantic. This can occur through an alignment of the subtropical jet with the African coastline and anomalous south-westerly surface to mid-tropospheric flow, leading to large-scale ascending motions and instability over the Western Mediterranean region (Dayan et al., 2015; Khouakhi et al., 2022; Toreti et al., 2010). These regional dynamical drivers of precipitation over Morocco have been studied in terms of both North Atlantic and Mediterranean circulation regimes (Driouech et al., 2010; Mastrantonas et al., 2020; Tramblay et al., 2012). While certain regimes over both regions, such as the negative phase of the





North Atlantic Oscillation (NAO), are associated with an increase in the probability of extreme precipitation, the dynamical mechanisms described above and analysed in terms of distinct patterns in Chaqdid et al. (2023) are not clearly captured in the regimes over either region. This motivates the application of the introduced machine learning method to this region, as a first case study of whether the approach is able to identify dynamical drivers that are informative of precipitation over Morocco but which also present an interpretable and predictable partitioning of the large-scale atmospheric phase space.

In terms of timescales, we choose to evaluate teleconnections and predictability at subseasonal to seasonal lead times. At these lead times, regional dynamical drivers of precipitation extremes in Morocco have been shown to be modulated by both the Madden-Julian Oscillation (MJO) (Gadouali et al., 2020) and variability of the northern-hemisphere stratospheric polar vortex (SPV) (Zhang et al., 2024). The MJO is a leading mode of global subseasonal variability that modulates deep tropical convection and thereby acts as a source of Rossby waves leading to teleconnections to extratropical regions (Lee et al., 2020; 85 Roundy et al., 2010). Variability in the SPV, on the other hand, has been shown to influence subseasonal forecast skill over the European and Mediterranean region, with weaker SPV states leading to an equatorward shift of the tropospheric eddy-driven jet and associated storm tracks (Kidston et al., 2015; Kretschmer et al., 2018).

The contribution of this paper is threefold. The first contribution is to introduce the CMM-VAE method, short for Categorical Mixture Model Variational Autoencoder, which enables the application of the method previously presented in Spuler et al. 90 (2024a) to categorical target variables. This allows us to inform the targeted circulation regimes with the spatial structure of precipitation extremes, working with a more realistic precipitation dataset (CHIRPS) instead of the reanalysis precipitation data used in Spuler et al. (2024a).

The second contribution is to use the CMM-VAE method to identify and analyse targeted regime representations of the dynamical drivers of extreme precipitation over Morocco. We compare the informativeness of these targeted regimes to con-95 ventional non-targeted circulation regimes identified using Principal Component Analysis (PCA) and k-means clustering, as well as targeted clusters identified using a linear targeted method, Canonical Correlation Analysis, again combined with k-means clustering.

The third contribution of the paper is to investigate the predictability of these targeted circulation regimes, compared to conventional, non-targeted, regimes. We first analyse the ability of subseasonal dynamical reforecasts to predict the targeted 100 regimes which provides an assessment of the predictability of the regimes conditional on state-of-the-art dynamical models. We then evaluate the predictability of the regimes in reanalysis data, conditional on teleconnections from two relevant modes of subseasonal variability, the MJO and SPV. We analyse conditional predictability in terms of the information-theoretical metrics of conditional entropy and mutual information. Next to presenting another line of evidence for the predictability of the regimes, this investigation shows whether the targeted CMM-VAE regimes can be interpreted as representing physical processes that 105 are modulated by large-scale drivers and can hence be used for further applications such as statistical downscaling as well as to improve the understanding of dynamical processes over the region.

The remainder of the paper is structured as follows. Section 2 introduces the CMM-VAE method for identifying targeted regimes (2.2), and presents the data and methods used to capture spatial patterns of extreme precipitation over Morocco (2.1), as well as to analyse the MJO and SPV teleconnections and skill in subseasonal dynamical reforecasts (2.3). Section 3 presents





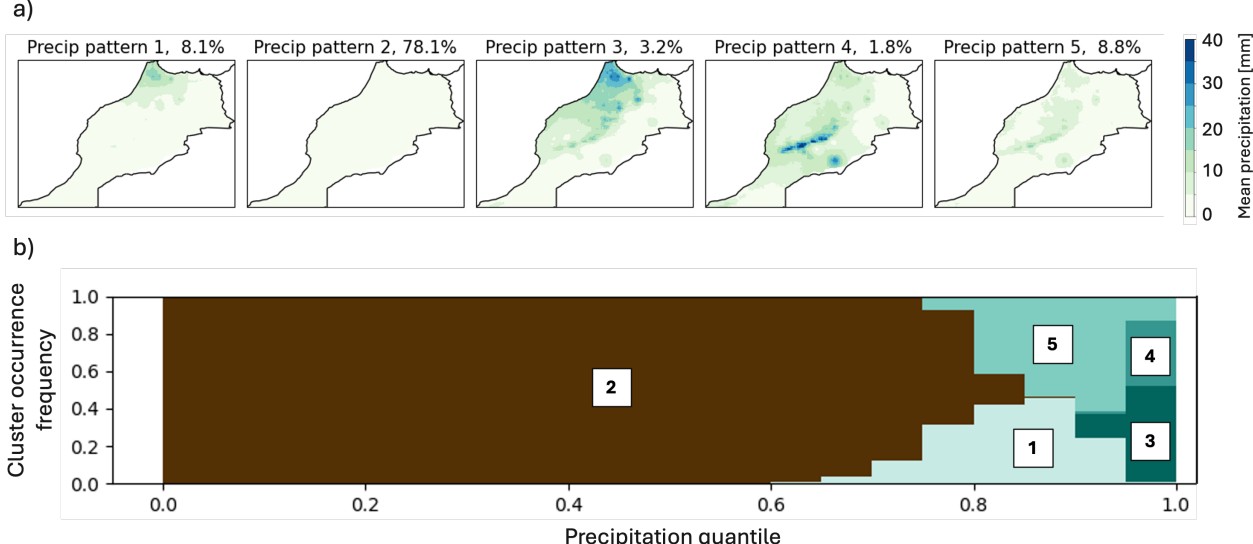

**Figure 2.** Top row: spatial patterns of 3-day precipitation events over Morocco in extended winter (November - March) identified using k-means clustering. The percentage number in the heading indicates the occurrence probability over all days. Bottom row: occurrence probability of clusters in different quantiles of total precipitation over Morocco.

the results of this study: the application of the CMM-VAE method to identify circulation regimes targeted to precipitation over Morocco (3.1), the evaluation of the forecast skill of these regimes in subseasonal hindcasts (3.2), as well as subseasonal teleconnections to the MJO and SPV in reanalysis data (3.3) Section 4 presents a discussion of the results, a conclusion and an outlook.

## 2 Data and Methods

### 2.1 Extreme precipitation over Morocco as local target variable

As target variable, we consider extreme precipitation over Morocco in extended winter (November-March). To this end, CHIRPS v2.0 (Funk et al., 2015) precipitation data is averaged over three days at each grid cell, as this timescale captures the duration of most extreme precipitation events over the observational period (Loudyi et al., 2022). The CHIRPS v2.0 dataset combines in-situ station data with satellite data for all longitudes and 50°S - 50°N and is available from 1981 to present at 0.05° resolution. The dataset was chosen as it has a more realistic representation of precipitation compared to the reanalysis data used in Spuler et al. (2024a) and shows good performance over Africa compared to other available gridded rainfall datasets (Dinku et al., 2018; Maidment et al., 2017).

To capture precipitation extremes, we analyse the 95th percentile at each grid cell as well as of spatially averaged precipitation over Morocco. Moreover, we compute the dominant spatial patterns of precipitation by applying k-means clustering on





the precipitation data described above, to be able to capture different dynamical drivers of precipitation over different regions (Chaqdid et al., 2023). Results for different choices of k were assessed and a cluster number of 5 was chosen as the minimal number that represents the most prevalent distinct spatial patterns of precipitation over the region. The resulting precipitation patterns shown in Figure 2 contain information about both common spatial patterns of precipitation (top row) as well as the extremality of these precipitation events (bottom row): Pattern 2 summarises all days associated with no or little precipitation,

while patterns 3 and 4, represent most days above the 95th percentile of total precipitation over Morocco. On the other hand, patterns 1 and 3 (and likewise 4 and 5) represent related spatial patterns but different levels of extremality.

## 2.2 Identifying (targeted) circulation regimes as regional dynamical drivers

Atmospheric circulation patterns are investigated using geopotential height data at 500 hPa (z500) over the East Atlantic and Mediterranean region (20°N - 80°N; 50°W - 30°E) in extended winter (November - March) based on ERA5 reanalysis data

from 1981 to 2022 (Hersbach et al., 2020) re-gridded to a resolution of 2.5° x 2.5°. The geopotential height data is standardized by subtracting the climatological daily mean and dividing the result by the standard deviation across grid points.

This choice of region was based on multiple considerations. Previous studies found the North Atlantic to be the key moisture source for precipitation over Morocco, and existing literature identifies the NAO as one of the dynamical drivers of precipitation over Morocco (Driouech et al., 2010; Khouakhi et al., 2022; Tramblay et al., 2012). Furthermore, we found that the anomalies

related to circulation regimes targeted to precipitation over Morocco over the Mediterranean region analysed in Spuler et al. (2024a) extend to the North Atlantic region. However, key results of this paper were found not to be sensitive to the choice of region.

Table 1 provides an overview of the different methods for identifying circulation regimes used in this study which are described in detail below.

### 2.2.1 Principal Component Analysis and k-means clustering (PCA + k-means)

Principal Component Analysis (PCA, commonly referred to as Empirical Orthogonal Function analysis in atmospheric science) combined with k-means clustering have established themselves as a common choice for determining non-targeted circulation regimes (Charlton-Perez et al., 2018; Michelangeli et al., 1995), including over the Atlantic and Mediterranean regions (Giuntoli et al., 2022; Mastrantonas et al., 2020). While other approaches exist (Hannachi et al., 2017), this two-step approach

(hereafter abbreviated PCA + k-means) is used as a baseline here against which to benchmark the targeted method. PCA is a linear dimensionality reduction method that projects a higher-dimensional input space into a reduced space spanned by the orthogonal eigenvectors of the covariance matrix of the data (e.g. Jolliffe and Cadima (2016)). K-means clustering is then applied to the reduced space of principal components and partitions the data into k sets that minimize the mean within-cluster squared distance from the respective cluster centre. PCA is implemented using the eofs package (Dawson, 2016) and the first

15 principal components are retained, while k-means clustering is applied using the Python sklearn implementation.





**Table 1.** Overview of methods used for identifying (targeted) circulation regimes.

| Abbreviation | Method | Target variable |
|---|---|---|
| PCA + k-means | Linear dimensionality reduction using Principal Component Analysis and subsequent k-means clustering | None |
| CCA + k-means | Linear dimensionality reduction using regularized Canonical Correlation Analysis and subsequent k-means clustering | CHIRPS precipitation over Morocco aggregated by sub-region |
| CMM-VAE | Categorical Mixture Model - Variational Autoencoder: nonlinear dimensionality reduction and probabilistic clustering | Spatial patterns of extreme precipitation over Morocco shown in Figure 2. |

#### 2.2.2 Regularized Canonical Correlation Analysis with k-means clustering (CCA + k-means)

Canonical Correlation Analysis (CCA) is a linear dimensionality reduction method that jointly identifies respective linear transformations of two high-dimensional spaces onto subspaces that maximize the correlation between the projections of the variables onto their new basis vectors (Johnson and Wichern, 2013). The method has previously been applied to identify tar-
geted circulation regimes (Vrac and Yiou, 2010), and is therefore implemented here, in combination with k-means clustering, to provide a well-established linear targeted method against which to compare methods based on nonlinear variational autoencoders. However, both the precipitation and geopotential height field were found to be too high dimensional for traditional CCA to algorithmically converge. We therefore aggregated precipitation at the district level over Morocco and implemented a ridge regularization parameter for the geopotential height field (Vinod, 1976). The regularized CCA is implemented using
the Python package cca-zoo (Chapman and Wang, 2021). Due to the aggregation of the precipitation field, only the first 10 canonical covariates could be computed and were used for subsequent clustering.

#### 2.2.3 CMM-VAE: a nonlinear targeted method based on variational autoencoders

We introduce a novel method, referred to as CMM-VAE, for identifying probabilistic and targeted circulation regimes based on a variational autoencoder, building on the RMM-VAE method previously introduced in Spuler et al. (2024a).
A variational autoencoder (VAE) is a deep generative machine learning method used for non-linear dimensionality reduction, that is to find a reduced representation of a high-dimensional input space. This reduced, also referred to as latent, space is identified by explicitly fitting a probability distribution, usually a multivariate Gaussian, using Bayesian variational inference (Kingma and Welling, 2013). The method is referred to as generative because the resulting reduced space is continuous and can therefore be used to generate new higher-dimensional samples.





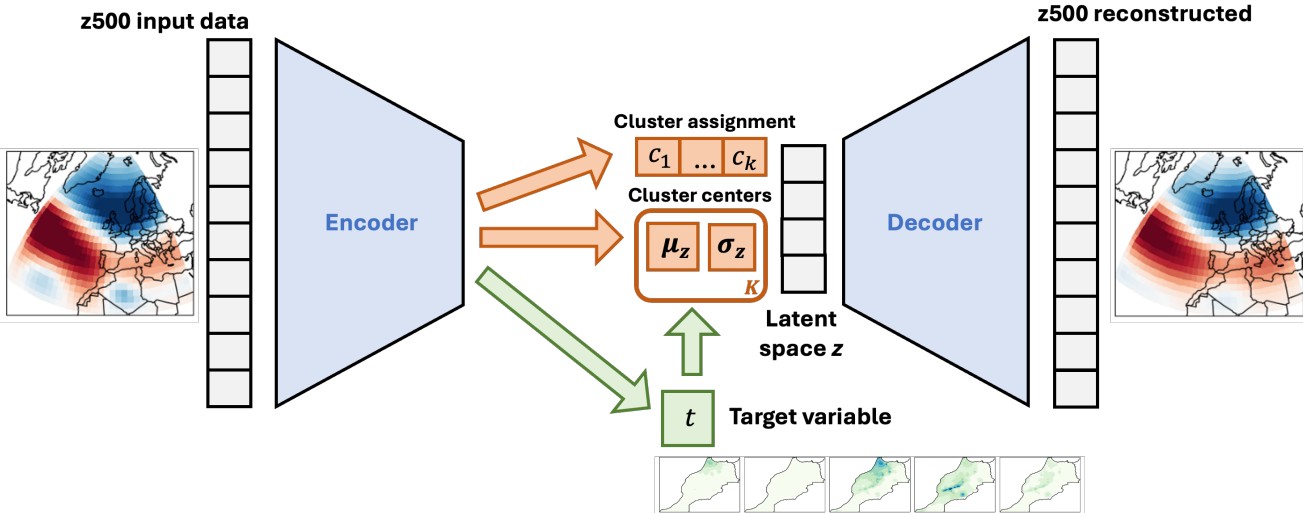

**Figure 3.** Graphical illustration of the variational autoencoder architecture underlying the CMM-VAE method. Normalized geopotential height data is input into the encoder which is a neural network of three dense layers of decreasing dimensionality. In the latent space, the method fits $k$ multivariate Gaussian distributions with means $\mu$ and standard deviations $\sigma$ and the cluster assignments $c_i$ of individual days (orange arrows), as well as a regression to the target variable $t$ which is used to regularize the latent space (green arrows). The decoder mirrors the encoder in architecture and reconstructs the original input data from the model fit in the latent space.

To identify targeted circulation regimes, Spuler et al. (2024a) extend the baseline VAE architecture in two ways to develop a method called Regression Mixture Model Variational Autoencoder (RMM-VAE). One is to fit a Gaussian mixture model (i.e. a mixture of several Gaussian distributions) into the reduced space to identify probabilistic circulation regimes, instead of a single multivariate Gaussian distribution. The method thereby combines dimensionality reduction and probabilistic clustering in a coherent statistical model (orange arrows in Figure 3). In contrast, other conventional methods for identifying weather regimes

implement dimensionality reduction and clustering separately and often conduct a 'hard', as opposed to a probabilistic, cluster assignment of individual days. The probabilistic cluster assignment implemented in the RMM-VAE method, retains more information on transitional states between clusters.

    The second modification introduced in the targeted VAE method is to use the encoder of the architecture to predict the chosen target variable, which, in our case, is precipitation over Morocco. The predicted target variable is then used to inform

(i.e. regularize) the latent space to obtain clusters that contain a more coherent response in terms of the target variable (green arrows in Figure 3). Spuler et al. (2024a) show that this regularization organises the latent space in terms of the target variable, that is disentangles the reduced dimension associated with changes in the target variable. This leads to the identification of circulation regimes that are more coherent in terms of their precipitation response. These modifications to the original VAE architecture are introduced by deriving a modified loss function of the architecture.





The CMM-VAE method extends the RMM-VAE method in the following way. The underlying loss function derived for the RMM-VAE method required the target variable to be a scalar Gaussian, which limits the applicability of the method. Here we derive a modification of the loss function which enables the application of the method to higher-dimensional categorical target variables and is therefore called CMM-VAE (Categorical Mixture Model - Variational Autoencoder). Instead of a linear regression in the prediction component of the encoder, the CMM-VAE method fits a higher-dimensional logistic regression.

Furthermore, the subsequent regularization of the latent space is modified. Instead of implementing the regularization as a regression from the continuous target variable to the latent space, which is how the loss function of the RMM-VAE is derived, the CMM-VAE method predicts the categorical cluster assignment from the (also categorical) target variable. This enables the regularization of the latent space using a categorical target variable. The loss function derived for the CMM-VAE architecture, as well as a more detailed explanation of differences to the RMM-VAE method, can be found in Appendix A.

The encoder and decoder of the CMM-VAE architecture were implemented using three dense neural network layers of decreasing dimensionalities (256, 128 and 64) and a ReLU activation function. The architecture was implemented using the Python library keras (Chollet et al., 2015). The models were iteratively trained for 150 epochs with a batch size of 128 and evaluated on different train-test splits in a k-fold cross-validation approach. The best-performing weights were then used to encode the entire dataset. A latent space of dimensionality 15 was selected.

## 2.3   Predictability metrics

### 2.3.1   The predictive skill of regimes in subseasonal hindcast experiments

The skill of subseasonal dynamical reforecasts in predicting the occurrence of the (targeted) circulation regimes is analysed using a lower-resolution reforecast experiment using the 47r3 cycle of the ECMWF IFS (CY47R3_LR) developed by Roberts et al. (2023) ranging back to 1980. This lower-resolution reforecast was shown to predict circulation regimes over the region
sufficiently well to justify the trade-off between lower resolution and extended time period.

    The 11-member ensemble forecasts of geopotential height at 500hPa up to lead times of 47 days, initialized on the 1st, 8th, 15th and 22nd of each month, were downloaded through MARS for the period 1980-2020. Start dates covering the extended winter period November - March were selected (i.e. from 22/09 to 22/03). The reforecasts were pre-processed to match ERA5 data: after selecting the region over which the reanalysis data was analyzed (20-80°N / 50°W-30°E) and re-gridding reforecasts
to a resolution of 2.5°x2.5°, the climatological mean was subtracted, and the result was divided by the standard deviation across grid cells. Both the mean and standard deviation were calculated for each day of the year and lead time independently across ensemble members. Finally, for each ensemble member, a rolling window mean of five days was calculated to correspond to the window length chosen in the reanalysis data.

    Following these pre-processing steps, the reforecasts were projected onto the circulation regimes calculated in the reanal-
ysis data. For the two linear methods, PCA and CCA, data for each ensemble member, lead time and initialization date was projected onto the principal component or canonical covariates pre-computed on reanalysis data. Subsequently, the k-means clustering fitted on reanalysis data was applied to the projected reforecasts to predict cluster assignments of reforecasts. For



the variational autoencoder method, the VAE trained on reanalysis data was used to predict the latent space, cluster assignment and reconstruction of reforecasts for each ensemble member, lead time and initialization date.

The forecast skill of the circulation regimes in subseasonal dynamical reforecasts was evaluated using (1) the Brier skill score extended to a multi-category forecast and (2) the area under the Receiver Operating Characteristic (ROC-AUC). The Brier score is a strictly proper scoring rule defined as $BS = \frac{1}{N}\sum_{n=1}^{N}\sum_{j=1}^{m}(\delta_{i_n j} - p_j)^2$ (Gneiting and Raftery, 2007), where $m$ is the number of forecast categories and $N$ is the number of timesteps. $\delta_{ij}$ is the Kronecker delta which equals 1 if the observation $i$ at timestep $n$ corresponds to category $j$, and 0 otherwise, and $p_j$ the forecast probability of category $j$. Based on

the Brier score, the Brier skill score was calculated with respect to the skill score of a climatological forecast in the following way: BSS = 1 - BS_forecast / BS_climatology. To compare the performance across methods, the score was calculated over all regimes since the predictive skill of individual regimes cannot be directly compared between methods due to the non-correspondence of individual regimes. The ROC curve shows the hit rate of the forecast over the false alarm rate as a function of the threshold (that a forecast must exceed to define a hit) extended to a multi-category forecast.

## 235 2.4 Conditional predictability in reanalysis data based on information theory

For the characterisation of the MJO, the real-time multivariate (RMM) MJO index was used (Wheeler and Hendon, 2004). This index is based on the first two principal components of combined fields of daily anomalies in 15°S-15°N outgoing longwave radiation, zonal winds at 850 and 200hPa, and the removal of the interannual variability by linear regression against the SST time series reflecting ENSO, which gives an RMM1 and RMM2 index. These two indices can be plotted in a phase diagram,

where an amplitude larger than 1 represents the occurrence of an MJO event, and the angle assigns a day to MJO phases 1 to 8, which reflect the propagation of the MJO from Africa over the Indian Ocean and the Maritime Continent to the Western Pacific. MJO indices were accessed from the Australian Bureau of Meteorology and calculated based on Gottschalck et al. (2010) and Wheeler and Hendon (2004).

To investigate the tropospheric impacts of weak and strong polar vortex states, the zonally averaged zonal wind at 60°N and

100 hPa was calculated from December to March and divided into terciles over the entire season to reflect weak, neutral and strong vortex conditions based on ERA5 reanalysis data. The pressure level of 100 hPa was chosen to capture the downward impact of the stratospheric variability. This variable and index has been used in previous studies, including Charlton-Perez et al. (2018).

Subseasonal teleconnections from both the MJO and SPV are themselves known to be modulated by seasonal modes of

variability such as the QBO and ENSO (Lee et al., 2019; Toms et al., 2020). While these are not directly investigated here, the seasonal intermittency of the subseasonal teleconnections is assessed using a block-bootstrapping approach which provides an estimate of the robustness of the subseasonal teleconnection across seasons (Roberts et al., 2023). Furthermore, the modulation of the SPV by the MJO (Garfinkel et al., 2014) is not assessed.

The predictability of the (targeted) regimes given these teleconnections was then evaluated based on information theory,

an approach which has been applied in climate science and machine learning (DelSole, 2004; Fang et al., 2024; Runge et al., 2012). In particular, the conditional entropy and mutual information between the regimes and the two subseasonal





teleconnections are evaluated (Murphy, 2022). Given two variables $X$ and $Y$, their individual entropies $H(X)$ and $H(Y)$ measure the average uncertainty inherent in the possible outcomes of the variable (Murphy, 2022), calculated as follows $H(X) = -\sum_{x \in X} p(x) \log p(x)$. Here, $Y$ is taken to be the regime and $X$ the phase or tercile of the large-scale driver (MJO

or SPV). Based on previous literature, we assume that knowing the phase or tercile of the large-scale driver (i.e. $X$) will give us information about, or reduce the uncertainty in Y. This can be formalized as the metric of conditional entropy $H(Y|X)$ that quantifies the amount of uncertainty remaining about the target variable $Y$ given that $X$ is observed, and therefore provides an estimate of conditional predictability. Given two discrete variables, their conditional entropy is calculated as follows: $H(Y|X) = \mathbb{E}_{p(X)}[H(p(Y|X)] = -\sum_x p(x) \sum_y p(y|x) \log p(y|x)$. Subtracting the conditional entropy from the uncertainty

inherent in the variable Y gives a symmetric measure of information shared between two variables, that is, their mutual information: $I(X,Y) = H(Y) - H(Y|X) = H(X) - H(X|Y)$. Here, mutual information is adjusted to account for the mutual information that would be detected between two independent sets of clusters (Vinh et al., 2010).

Conditional entropy provides an estimate of the average conditional predictability of the regional dynamical drivers given the large-scale teleconnections over lead times, irrespective of the specific model chosen to make this prediction. However, under

distributional assumptions, conditional entropy has been shown to provide a lower bound to mean squared error achievable in a regression model aiming to predict $Y$ from $X$ (Vinh et al., 2010). Since conditional entropy does not take into account the uncertainty in the atmospheric regimes themselves, both conditional entropy and adjusted mutual information were evaluated. In contrast to metrics such as momentary information transfer proposed by Runge et al. (2012), evaluating mutual information and conditional entropy here does not disentangle the information transfer from $X$ to $Y$ at a specific lead time $\tau$ from the

information transfer at lead time $\tau - \epsilon$ combined with autocorrelation of $Y$ during the time $\epsilon$. However, disentangling this difference is not considered relevant for our present purpose.

## 3   Results

### 3.1   Characteristics of the circulation regimes and their informativeness precipitation over Morocco

Figure 4 shows the circulation regimes identified by the different methods, alongside the odds ratio of extreme precipitation at

each grid cell during the days assigned to the respective regimes.

The PCA + k-means method identifies the regimes expected from previous literature: the two phases of the NAO (regimes 1 and 3), Scandinavian Blocking (regime 2), an Atlantic low and the Atlantic Ridge (regimes 4 and 5) (Falkena et al., 2020; Michelangeli et al., 1995). The negative phase of the NAO is associated with a moderate increase in the probability of extreme precipitation, which is in line with existing literature that finds a correlation between the NAO- and wet conditions over

Morocco due to a southward shift of the North Atlantic storm tracks (Driouech et al., 2021; Tramblay et al., 2012).

The CMM-VAE method also identifies the negative phase of the NAO and finds it to be associated with a moderate increase in extreme precipitation (CMM-VAE regime 2). However, CMM-VAE identifies another dynamical pattern associated with an even higher increase of extreme precipitation over Morocco (CMM-VAE regime 1): this regime is related to a Scandinavian Blocking alongside a localized low around the western coast of the Iberian Peninsula and Morocco. The dynamical pattern



**Figure 4.** Identified circulation regimes (top rows) and corresponding odds ratios of extreme precipitation (bottom rows) for the four different methods with the number of clusters specified as $k = 5$. The regime frequencies are given in percent. The odds ratio of extreme precipitation corresponds to the ratio of the probability of the climatological 95th percentile of precipitation at the grid cell conditional on that circulation regime, divided by the unconditional probability of 95th percentile of precipitation (i.e. 0.05). The regimes are ordered in decreasing order of total precipitation during the days assigned to this cluster by the respective method.




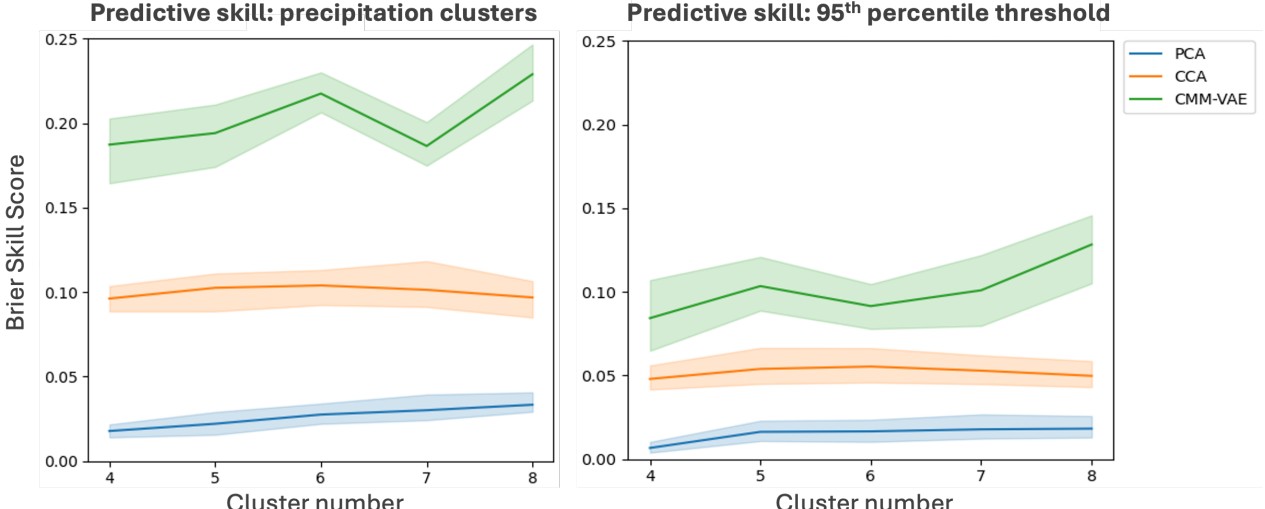

**Figure 5.** Informativeness of the regimes of exceedance of the 95th percentile of total precipitation over Morocco (right) and precipitation clusters shown in Figure 2 (left), evaluated using the Brier Skill Score. 95% confidence interval computed based on bootstrap procedure with n=50.

represented by this additional regime is consistent with the geopotential height anomalies during extreme precipitation events analysed in previous publications (Chaqdid et al., 2023; Toreti et al., 2010). Dynamically, it relates to (south-)westerly mid-tropospheric flow and associated moisture transport from the Atlantic found to drive precipitation over Morocco (Dayan et al., 2015; Khouakhi et al., 2022). The associated low-level zonal wind and streamfunction anomalies shown in Appendix B indicate a split jet configuration. In contrast, the non-targeted PCA + k-means regimes do not show this additional pattern and do not resolve an increase in extreme precipitation associated with the Scandinavian Blocking regime as it lacks the resolution of the localized low off the coast of the Iberian Peninsula.

Aside from disentangling this additional regime modulating extreme precipitation over Morocco, the CMM-VAE method identifies regimes similar to those found in the non-targeted PCA + k-means clustering approach: the NAO+ and Atlantic Ridge regimes look relatively similar, while the CMM-VAE method identifies a slightly southward shifted Scandinavian Blocking regime that is associated with a positive geopotential height anomaly over the entire Mediterranean region. The targeted regimes are statistically well separated and show an overall only slightly reduced persistence compared to the non-targeted regimes (see Appendix B).

The CCA + k-means method projects the geopotential height data into a subspace which is maximally correlated with precipitation over Morocco. It identifies targeted clusters which are associated with an increase in extreme precipitation over Morocco but which show little other structure in the atmospheric phase space.

Overall, we find that the CMM-VAE is able to identify a regime representation of dynamical conditions over the region that is more informative of precipitation extremes over Morocco by disentangling a dynamical driver not identified in conventional





PCA + k-means regimes. In contrast to CCA, the CMM-VAE method identifies more structure overall in the atmospheric phase space, and therefore regimes which are more persistent and statistically robust (see Appendix B).

To further quantify the informativeness of the regimes of precipitation over Morocco, we construct a forecast of extreme precipitation and precipitation clusters based on the regime occurrence and the associated conditional probability of the target variable. We compare the skill of this forecast for the different methods (Figure 5). The higher the skill, the stronger the link between circulation regimes and precipitation over Morocco.

We find that the CMM-VAE method outperforms PCA + k-means and CCA + k-means in terms of predicting both the
precipitation clusters (Figure 5, left) as well as the exceedance of 95th percentile precipitation (Figure 5, right), hence identifying circulation regimes that are more informative of the extreme precipitation over Morocco and confirming the analysis of odds ratios shown in Figure 4. The skill is overall higher for predicting the precipitation cluster assignment compared to the threshold exceedance of 95th percentile precipitation.

The cluster number for further investigation, $k = 5$, was selected on the basis of the robustness of cluster centers to subsam-
pling analyzed in Spuler et al. (2024a). In sensitivity checks performed, it was found that the principle results presented in this paper are not sensitive to the choice of cluster number.

## 3.2   How well are the circulation regimes predicted in subseasonal hindcasts?

We now investigate the skill of dynamical subseasonal hindcasts in predicting the circulation regimes. The previous section showed that the CMM-VAE method identifies regimes that are more informative of the local-scale impact, namely extreme
precipitation over Morocco. The aim of the analysis performed in this section is to investigate if the CMM-VAE regimes are as predictable as those identified with the conventional PCA + k-means approach. Together with the enhanced informativeness of the target variable, this would provide evidence that the CMM-VAE method can identify more suitable representations of regional dynamical drivers that can help improve predictions of precipitation over Morocco across a range of timescales.

The forecast skill of the different regimes in subseasonal hindcasts is assessed using two evaluation metrics: the Brier Skill
Score (BSS) and the Area Under the Curve of the Receiver Operator Characteristic (ROC-AUC). The ROC-AUC, shows the hit rate of the forecast over the false alarm rate as a function of a threshold extended to a multi-category forecast and has a similar interpretation to the resolution term of the BSS. Results are shown in Figure 6.

The targeted CMM-VAE regimes are found to be as predictable in terms of both BSS and ROC-AUC as the non-targeted regimes identified using PCA + k-means, and even appear to be predictable at slightly extended lead times: the BSS drops
below zero, that is climatological skill, after 19 days (CMM-VAE), 17 days (PCA) and 11 days (CCA) respectively. The CCA + k-means regimes are overall less predictable in both metrics. Skill drops below zero, i.e. below climatological skill, due to the imperfect climatological calibration of the reforecasts in all methods.

The BSS can be further decomposed into terms representing the reliability, i.e. calibration or conditional bias, the resolution of the forecast, and the observational uncertainty (Stephenson et al., 2008). We analyse this decomposition for the different
regimes to understand the similar performance of CMM-VAE and PCA + k-means regimes in terms of overall skill score (see



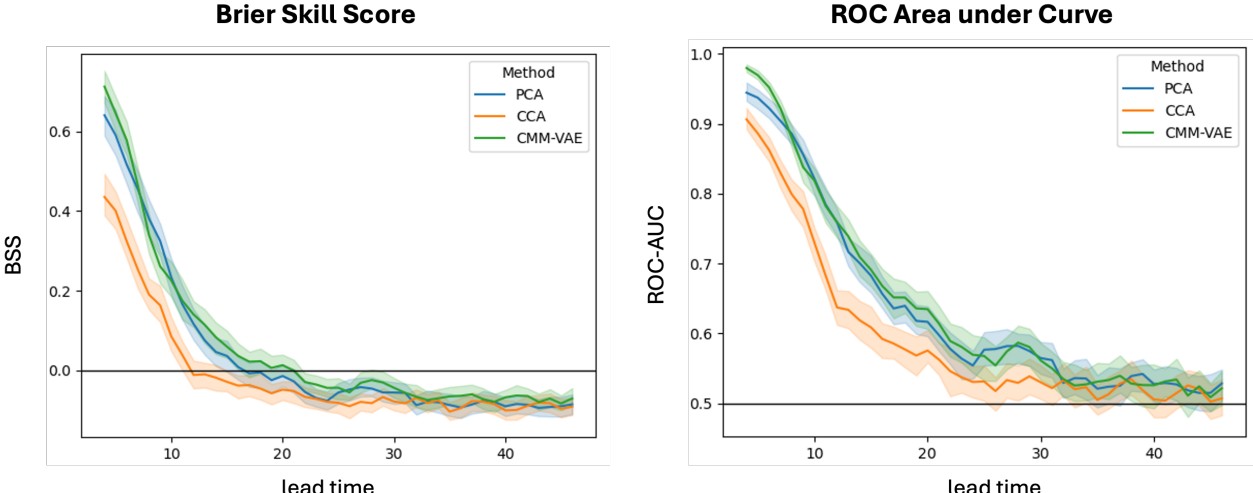

**Figure 6.** Brier Skill Score (left) and ROC AUC for circulation regime assignment predicted by the subseasonal hindcasts. Confidence interval based on a bootstrapping procedure with n=100.

Appendix C). We find that the CMM-VAE regimes perform slightly worse in terms of resolution but slightly better than PCA + k-means regimes in terms of reliability.

### 3.3 Teleconnections between subseasonal modes of variability and circulation regimes in reanalysis data

In this section, we investigate the predictability of the targeted regimes in reanalysis data, given two known subseasonal
teleconnections relevant to the region: the MJO and variability in the SPV. Predictability here is understood in the information theoretical sense as the amount of information shared between two sets of variables - subseasonal modes of variability and the targeted circulation regimes. This is assessed by first analyzing changes in the conditional probabilities of regime occurrence, and then building on this to evaluate adjusted mutual information and conditional entropy of the two sets of variables as information theoretical measures of predictability. This analysis provides insight into whether the predictability of the targeted
circulation regimes in subseasonal reforecasts is also physically interpretable in terms of large-scale dynamical drivers.

#### 3.3.1 Teleconnections from the stratospheric polar vortex

Figure 7 shows the change in the probability of the different circulation regimes, following weak, neutral or strong states of the polar vortex (labeled -1, 0 and 1 respectively). The circulation regimes are ordered as in Figure 4, i.e. by the occurrence of precipitation in each regime, from high to low.
We find that the influence of the SPV on precipitation over Morocco appears to be primarily modulated via the NAO with an increase in the probability of the NAO- (PCA regime 1 and CMM-VAE regime 2) following weak SPV states. On the other hand, the conditional probability of European blocking and the Atlantic Ridge regime (CMM-VAE regimes 3 and 5)





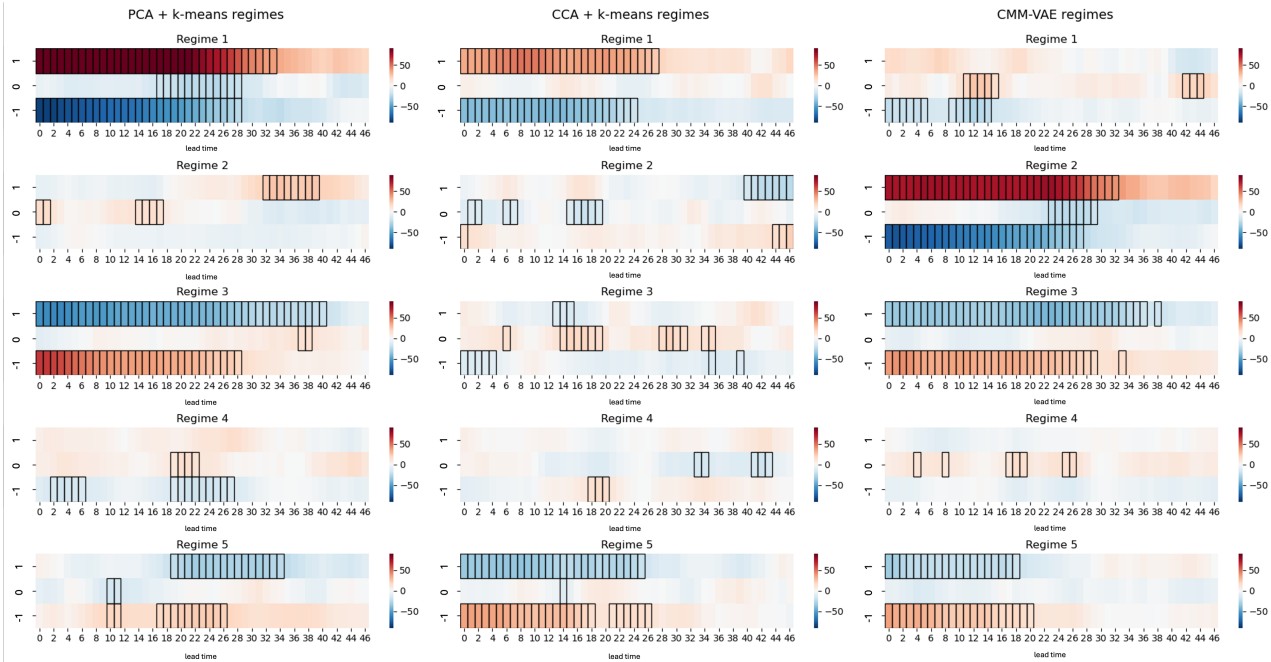

**Figure 7.** Change in conditional probability of the different circulation regimes (= absolute difference between conditional and unconditional probability, i.e. if a regime occurs around 10% of days and shows a 10% increase here, it becomes twice as likely) given or following weak (-1), neutral (0) and strong (1) states of the stratospheric polar vortex for lags up to 47 days. Statistically significant changes in the conditional probability are indicated using a black rectangle around the cell. These are calculated using a block-bootstrapping approach that samples entire DJFM seasons from the data with n=1000.

is reduced following weak vortex states. This result is in line with established findings on weaker SPV states leading to an equatorward shift of the tropospheric eddy-driven jet and associated storm tracks (Kidston et al., 2015; Kretschmer et al., 2018). The localized low associated with a Scandinavian Blocking (CMM-VAE regime 1), on the other hand, does not appear to be significantly modulated by the SPV. The CCA + k-means regimes to be less strongly modulated by the SPV, although we find a slight increase in the probability of the regime associated with the strongest increase in extreme precipitation following weak vortex states which is in line with the dynamical mechanisms found for the PCA + k-means and CMM-VAE regimes.

To quantify this difference in conditional predictability of the regimes given teleconnection from the SPV, we analyze the mutual information between the different regimes and SPV states, as well as the conditional entropy of the regimes given the SPV state. These metrics provide an information theoretical assessment of predictability that is here aggregated across regimes. Results are shown in Figure 8.

We find that conditional entropy, which quantifies the average uncertainty in the targeted circulation regime remaining given the state of the polar vortex is known, is similar for the PCA + k-means and CMM-VAE regimes and higher for the CCA





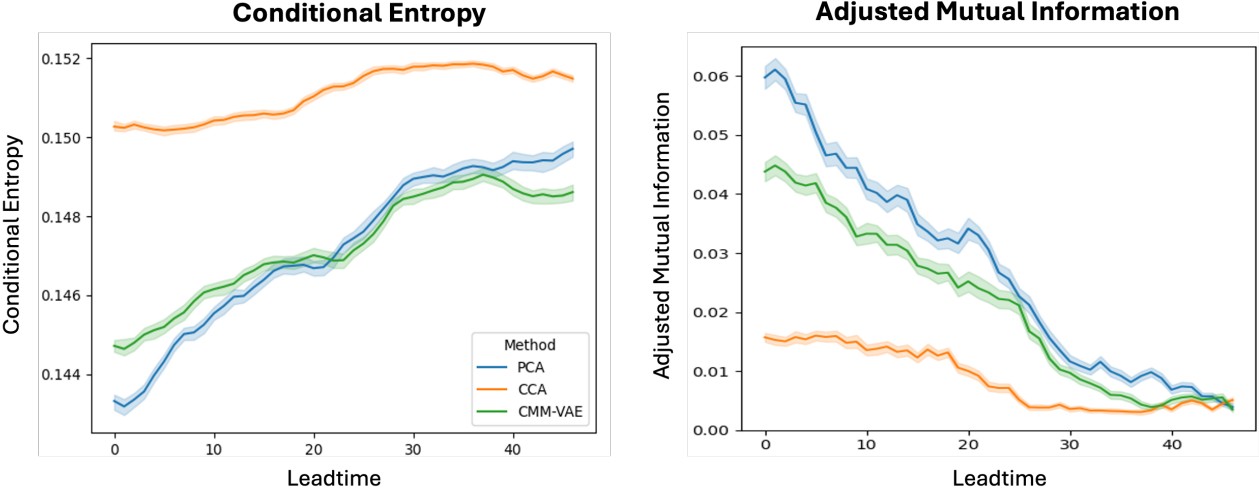

**Figure 8.** Left: Conditional entropy, i.e. the average uncertainty in the circulation regime given that the state of the polar vortex is known, averaged across the regimes. Lower values are better. Right: Adjusted Mutual Information. i.e. shared information between the circulation regime and the state of the stratospheric polar vortex which in addition to conditional entropy also accounts for the uncertainty in the regimes themselves, averaged across regimes. Higher values are better.

regimes. Mutual information between the regimes and SPV states, which also takes into account the uncertainty in the regimes themselves, is highest for the PCA + k-means regimes and lowest for the CCA regimes.

These results show that the CMM-VAE and PCA + k-means regimes are more predictable given knowledge of the SPV compared to the CCA + k-means regimes, hence capturing the downward impact of this stratospheric teleconnection better. The downward impact of the SPV assessed by these two metrics is found to decrease somewhat monotonically over time.

**3.3.2   Teleconnections from the Madden-Julian oscillation**

Figure 9 shows the change in probability of the different circulation regimes, following different phases of the MJO. The results highlight the oscillatory nature of the MJO, with the impact of different MJO phases on individual regimes propagating over lead times. Furthermore, large and significant changes in the conditional probabilities of individual regimes are found even at lead times up to 47 days, whereas stratospheric impacts tend to decay earlier (see Figure 7).

CMM-VAE regime 2 and PCA regime 1 which are associated with a negative NAO pattern, are modulated most strongly by the MJO, with a decrease in occurrence probability following MJO phases 1-4 and an increase following phases 6-8. This is consistent teleconnections between the MJO and NAO reported in the literature on the North Atlantic region (Cassou, 2008), as well as Morocco specifically (Gadouali et al., 2020). CMM-VAE regime 1, which is associated with the highest increase in the probability of extreme precipitation over Morocco, shows some modulation by the MJO.





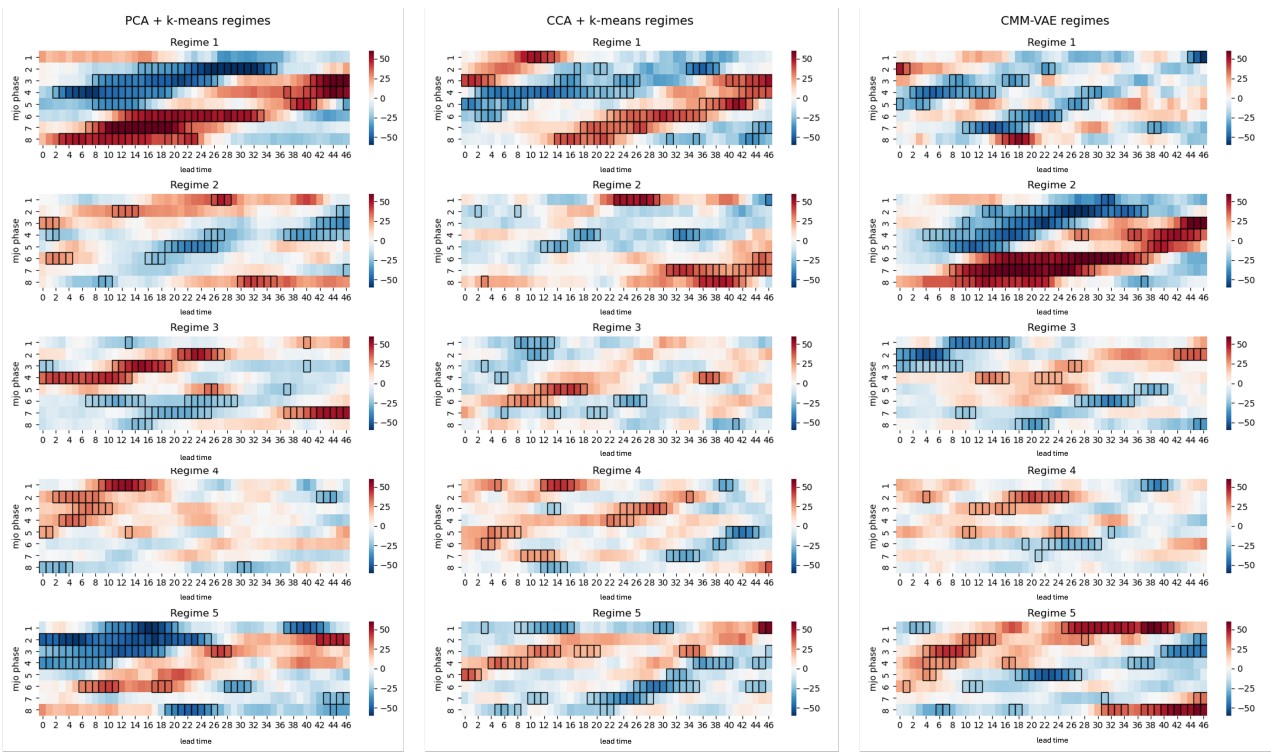

**Figure 9.** As in Figure 8 but for MJO phases.

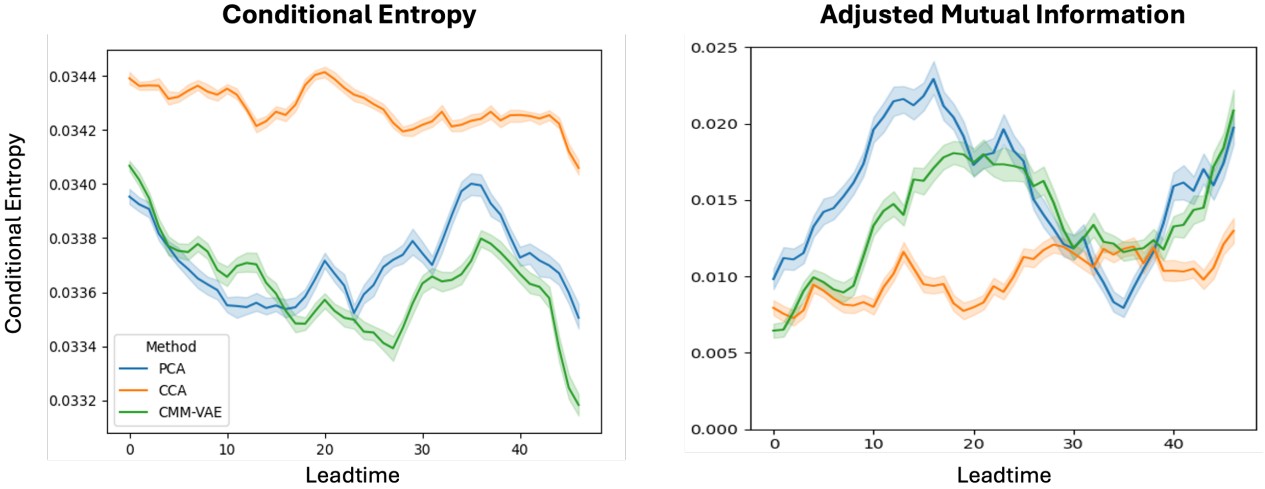

**Figure 10.** As in Figure 7 but for MJO phases.



Results for the conditional entropy and adjusted mutual information shown in Figure 10 highlight that the PCA + k-means and CMM-VAE regimes capture the dynamical teleconnection mechanisms between MJO and circulation over the Mediterranean region slightly better than CCA + k-means regimes, and hence are more predictable given knowledge of the MJO phase. In contrast to the teleconnection from the SPV, predictability from the MJO oscillates over lead times, and for the CMM-VAE regimes shows the lowest level of conditional entropy (highest level of mutual information) for lead times of 47 days.

## 4   Discussion and Conclusions

This paper introduces a novel method, the Categorical Mixture Model Variational Autoencoder (CMM-VAE) to identify regional dynamical drivers of a chosen impact variable in the form of targeted circulation regimes. Compared to two well-established linear methods for identifying circulation regimes, we find the targeted CMM-VAE regimes are more informative of the impact variable of interest while maintaining their predictability in subseasonal hindcasts and dynamical interpretability.

Applying the method to study drivers of precipitation over Morocco, we find that the method is able to disentangle an additional circulation pattern as a dynamical driver of extreme precipitation.

Through a regularized variational autoencoder architecture and modified loss function, the CMM-VAE method extends the method previously presented in Spuler et al. (2024a) to a higher-dimensional categorical target variable. This new method enables the identification of probabilistic circulation regimes targeted to spatial patterns of extreme precipitation over Morocco.

The identified regimes are compared to regimes identified using Principal Component Analysis and k-means clustering (PCA + k-means) as a baseline non-targeted method, and Canonical Correlation Analysis and k-means clustering (CCA + k-means) as a linear targeted clustering method.

The CMM-VAE method identifies a probabilistic partitioning of the atmospheric phase space that better disentangles dynamical patterns modulating extreme precipitation over Morocco (Figure 4), thereby enhancing the informativeness of the

resulting regimes (Figure 5). The additional regime identified by the CMM-VAE method, which is not found in the PCA or CCA + k-means regimes, is associated with a Scandinavian blocking together with a localised cut-off low off the coast of Morocco. This dynamical pattern is consistent with previous literature investigating dynamical drivers of extreme precipitation over Morocco and the Western Mediterranean (Chaqdid et al., 2023; Toreti et al., 2010).

Investigating the skill of dynamical subseasonal hindcasts in predicting the circulation regimes, we find that the targeted

CMM-VAE regimes are as predictable as the baseline non-targeted PCA + k-means regimes in subseasonal hindcasts, and more predictable than the regimes identified using CCA + k-means (Figure 6). This is a significant result compared to previous studies, which showed a trade-off between identifying locally informative patterns and regimes that are predictable at subseasonal lead times (Bloomfield et al., 2021). The result implies that in this region, the CMM-VAE method is able to identify a representation of regional dynamical drivers that balances and even resolves the trade-off between informativeness of local

impacts and subseasonal predictability.

The ability of this probabilistic machine learning method to strike a balance between local informativeness and predictability of the targeted regimes can be attributed to several factors. One is the efficiency of neural networks in identifying a non-





linear transformation function that encodes the information in a more informative reduced space and therefore enables the subsequent regularisation, i.e. targeting, of the dimensionality reduction. The second is that the loss function derived using

variational inference represents the different objectives of targeted clustering - such as representation of the full phase space and informativeness of the target variable - in a coherent statistical model that can be jointly optimized.

All methods for identifying regimes studied here require choosing the region over which to cluster atmospheric circulation, as well as the number of clusters $k$, a priori. The North Atlantic region was chosen based on previous literature highlighting the importance of dynamical drivers from the North Atlantic, but regimes were also analysed for circulation anomalies over

a smaller Mediterranean region. The number of clusters was chosen based on the sensitivity of the cluster centres to sub-sampling analysed in Spuler et al. (2024a), but results for other cluster numbers were also computed (e.g. Figure 5. We find that the improved informativeness of the CMM-VAE regimes, as well as their equal predictability in subseasonal hindcasts, are robust to both the choice of k as well as the choice of region.

We also investigate and explain this predictability in terms of subseasonal teleconnections relevant to the region, the MJO

and variability in the SPV. Conditional predictability given these teleconnections is analysed based on mutual information and conditional entropy, two information-theoretical measures of predictability. In line with the analysis of predictive skill in subseasonal hindcast data, the CMM-VAE regimes show similar levels of conditional predictability as the non-targeted PCA + k-means regimes. Conditional predictability of the regimes given the SPV is higher during strong or weak, as opposed to neutral, vortex states, and decays over subseasonal lead times. The conditional predictability of the regimes given the MJO, on

the other hand, shows a clear oscillation across subseasonal lead times. This result highlights potential windows of opportunity for subseasonal forecast skill in predicting precipitation extremes over Morocco.

Furthermore, the regimes disentangle distinct dynamical mechanisms through which extreme precipitation over Morocco is modulated by the MJO and variability in the SPV. The results suggest that the impact of both the MJO and SPV on precipitation over Morocco is mediated primarily via the NAO, while the CMM-VAE regime associated with a localised geopotential low

pattern and Scandinavian Blocking, which is the one associated with the highest increase in the probability of extreme precipitation, does not show a strong link to the SPV and is somewhat modulated by the MJO. This result highlights that the targeted CMM-VAE regimes - which are statistically optimised based on the local-scale variable - also represent physical processes that are modulated by large-scale drivers and can be used to understand the modulation of the frequency of precipitation extremes over Morocco by low-frequency modes of internal variability in the climate system.

While the focus of this paper is on predictability and teleconnection relationships at subseasonal lead times, these findings are relevant to seasonal timescales and studies of regional climate change. In subsequent work, targeted regimes could be investigated in future climate projections and used to condition bias adjustment and downscaling approaches (Dorrington et al., 2022; Maraun et al., 2010; Spuler et al., 2024b). The method could be further tested and refined through application to further regions and target variables. Finally, the conditional predictability of the regimes given the MJO and SPV could be

investigated in hindcast data and used to identify windows of forecast opportunity for extreme precipitation over Morocco.



*Code and data availability.* All data used in this study is stored and available for download in Zenodo: https://zenodo.org/records/14534652. The code will be made publicly available upon final publication.

*Author contributions.* Conceptualization: F.S., M.K., T.S., M.B. Methodology: F.S., M.K. Investigation, software, data curation: F.S. Visualisation: F.S., M.K. Supervision: M.K., T.S. Writing original draft: F.S., M.K., T.S; Writing review and editing: F.S., M.K., T.S., Y.K., M.B.

All authors approved the final submitted draft.

*Competing interests.* The authors declare no competing interests exist.

*Acknowledgements.* The authors thank Jakob Wessel for useful discussions and feedback. ERA5 reanalysis data (Hersbach et al., 2020) was downloaded from the Copernicus Climate Change Service (C3S) (2023). The results contain modified Copernicus Climate Change Service information 2020. Neither the European Commission nor ECMWF is responsible for any use that may be made of the Copernicus

information or data it contains. Subseasonal hindcasts were accessed through MARS, the ECMWF meteorological archive. The authors thank Chris Roberts for guidance on accessing the subseasonal hindcasts through MARS.



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





**Appendix A: Loss function derivation for the CMM-VAE method**

The two regularized methods, the Regression-Mixture Model Variational Autoencoder (RMM-VAE), introduced in Spuler et al. (2024a), and the CMM-VAE method introduced in this paper differ with respect to the way in which the latent space is regularized using the target variable. While in the RMM-VAE architecture, the target variable $t$ is used to directly regularize the latent space $z$ (central panel in Figure A1), this is not possible when working with higher dimensional categorical target variables. In the CMM-VAE architecture, we therefore instead regularize the cluster assignment $c$ using the target variable $t$. This is visualised in the graphical model shown in the right panel of Figure A1.

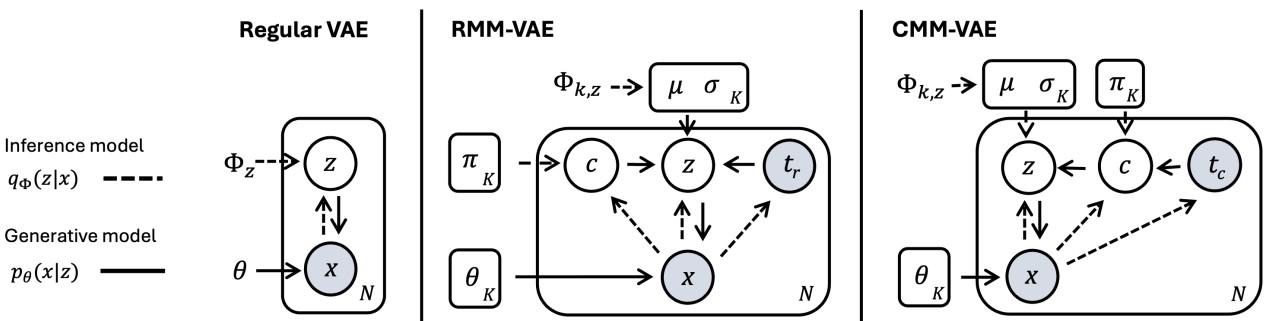

**Figure A1.** Representation of the statistical models underlying a regular variational autoencoder as graphical models used to derive the loss functions for the considered architectures, RMM-VAE and CMM-VAE. $x$ represents the input z500 space, $z$ - the identified latent space, $c$ - the cluster assignments of individual data points, and $t$ - the target variable. $\mu$ and $\sigma$ are the parameters of the Gaussian distributions fitted into the latent space, $\pi_k$ is the prior on the cluster occurrence frequency and $\theta_k$ are the parameters of the non-linear decoder.

This graphical model corresponds to the following decompositions of the inference and generative distributions.

$$q(z,c,t|x) = q(z|x) * q(c|x) * q(t|x) \text{ and } p(z,x,c,t) = p(x|z) * p(z|c) * p(c|t) * p(t) \tag{A1}$$

With these decompositions, we can now follow the standard procedure for Bayesian variational inference to derive the following loss function for the CMM-VAE architecture:

$$\mathcal{L}(x) = -D_{KL}(q_\phi(z,c,t \mid x) \mid p_\theta(x,z,t,c))$$

$$= \sum_k q_\phi(c^k \mid x) \left[ \mathbb{E}_{q_\phi(z|x)}[\log p_\theta(x \mid z)] - \mathbb{E}_{q_\phi(z|x)}[D_{KL}(q_\phi(z \mid x) \mid p(z \mid c^k))] \right.$$

$$\left. - \mathbb{E}_{q_\phi(t|x)}[D_{KL}(q_\phi(c^k \mid x) \mid p_\theta(c^k \mid t))] \right] - D_{KL}(q_\phi(t \mid x) \mid p(t)). \tag{A2}$$

For a more detailed explanation of variational autoencoders and variational inference applied to study atmospheric circulation and the way in which the regularization acts on the latent space, we refer to Spuler et al. (2024a), where this type of architecture is first used to study target circulation regimes.





## Appendix B: Further evaluation of the dimensionality reduction and statistical robustness of (targeted) regimes

In terms of reconstructing the original input space (Figure C1, left), the CMM-VAE method performs best by far, while CCA
performs worst. This finding is in line with the results presented in Spuler et al. (2024a) for the RMM-VAE method.

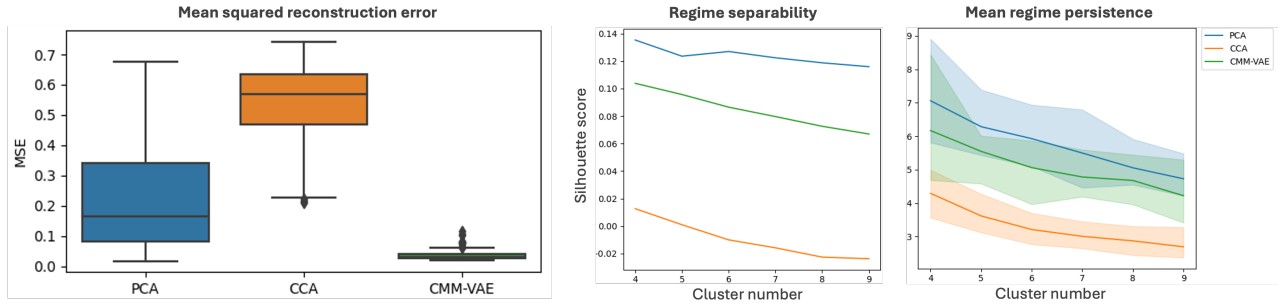

**Figure C1.** Local properties of the circulation regimes: (left) mean squared error between the z500 field reconstructed from the latent space
and the original z500 input field for k=8 regimes. Little sensitivity of the result to the choice of cluster number detected: (centre) silhouette
score of the clusters evaluated for different cluster numbers k; (right) mean regime persistence across the k regimes for different choices of
cluster number k.

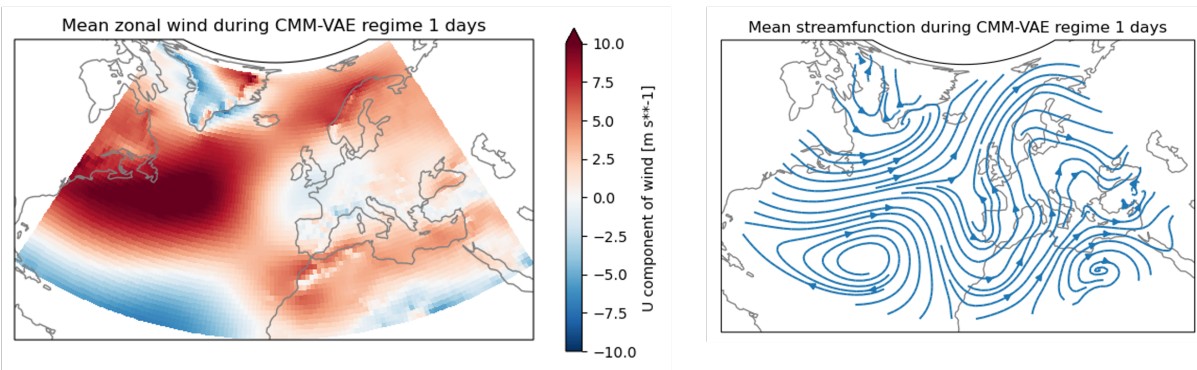

**Figure D1.** Mean anomalies in the zonal wind at 850hPa (left) and streamfunction calculated from zonal and meridional wind at 850hPa
(right) during days associated with CMM-VAE regime 1.

In terms of the regime separability and persistence (Figure C1, center and right), the non-targeted PCA + k-means method
performs best CCA performs worst according to both metrics, with a silhouette score being around 0, indicating overlapping
and statistically non-robust clusters, and the lowest regime persistence. The CMM-VAE method show a similar regime persis-
tence compared to PCA in some regimes, and a slightly lower persistence in others, indicating that the underlying dynamical
processes modulating the target variable are not as persistent. In terms of regime separability, The CMM-VAE method shows
a slightly lower silhouette score compared to the PCA + k-means method. However, a slight reduction would be expected



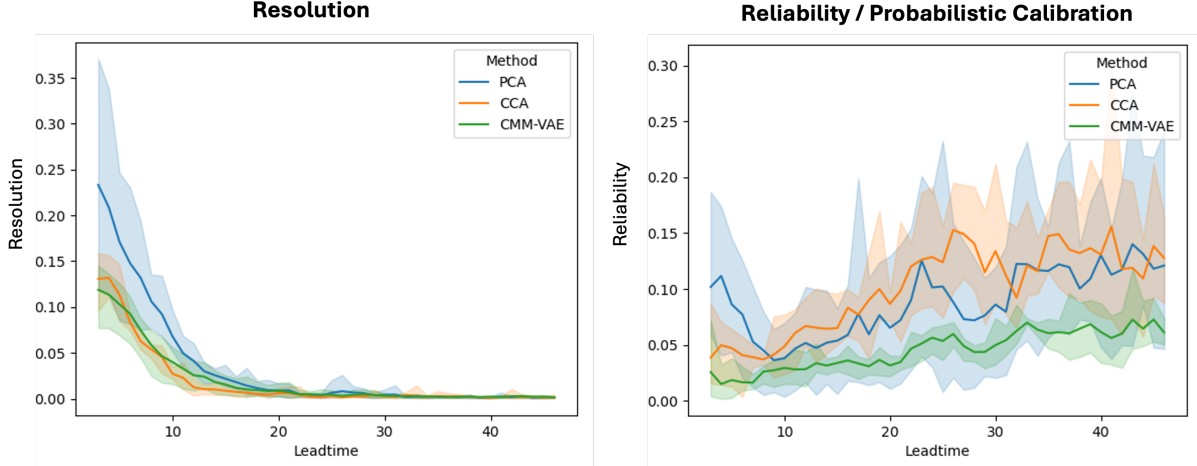

**Figure F1.** Decomposition of the Brier Skill Score in terms of resolution and reliability as described in the text above.

from probabilistic clusters as the silhouette score does not consider regime probabilities and is calculated on the basis of the assignment of a data point to the most likely cluster.

Overall, these results are found to be consistent with the results presented for the RMM-VAE method in Spuler et al. (2024a)) and are therefore included in the appendix here for completeness purposes.

## Appendix E: Extended forecast evaluation

The Brier Score can be decomposed into terms representing the reliability / probabilistic calibration, resolution and observational uncertainty of the forecast (Wilks 2019):

$$BS = \frac{1}{n}\sum_{i=1}^{I} N_i(y_i - \bar{o}_i)^2 - \frac{1}{n}\sum_{i=1}^{I} N_i(\bar{o}_i - \bar{o})^2 + \bar{o}(1 - \bar{o})^2 \tag{E1}$$

Computing this analytical decomposition requires binning the forecast probabilities (which calculating the Brier score does not), which makes the results more unstable than the actual score. The results for n=12 bins are shown in Figure F1. The reliability represents the squared difference from the diagonal and assesses the probabilistic calibration, or conditional bias, of the forecast (lower is better), while the resolution term represents the difference from the climatological occurrence frequency of the regime - the larger the difference of the forecast to a climatological forecast, the higher the resolution term.

We find that CMM-VAE performs worse than PCA in terms of resolution but better than PCA in terms of reliability. An interesting difference between PCA and CMM-VAE methods is the non-probabilistic vs probabilistic cluster assignment. This means that the two methods are affected differently by the choice of binning and, therefore, the overall skill score shown in Figure 6 might be the more robust metric to use here. Second, the forecasts in this study are evaluated against a single 'true'



observed cluster (for the CMM-VAE method, this corresponds to the cluster that is assigned the highest probability), which

705  might represent a disadvantage for the resolution of the CMM-VAE method.