# Peer review of "Learning predictable and informative dynamical drivers of extreme precipitation using variational autoencoders"

_EGUsphere, 2024_

## Referee Comment (RC1)

In this paper the authors address the question of how to identify useful weather regimes, which are both predictable in current forecasting systems and which do a good job at explaining variability in a target field of interest: in this case extreme wintertime rainfall over Morocco. Much ink has been spilled in the past 20 years on the topic of which weather regimes to use, how to define them, etc. I find the approach the authors have taken here refreshing and exciting. The use of a machine learning approach based on a variational autoencoder architecture to perform this task is, as far as I am aware, a novel innovation in this area, putting aside the authors' recent related paper. The focus on the practical value of the regimes for forecasting is also conceptually clarifying and provides a good motivation for the work.

The authors compare their CMM-VAE regimes to both a classical K-means approach and a surface-weather targeted CCA approach, and show that their method explains the most categorical rainfall variability while remaining similarly well predicted as the classical regime, with only a slight decrease in persistence and separability. They then go on to investigate sources of predictability, linking the predictability of the regime patterns to the polar vortex and Madden Julian oscillation.

**I really like this work, and my only comments are editorial in nature; I believe the paper is publishable after minor revisions to increase clarity and to fix a few small typos.**

Josh Dorrington

Minor points:
L55: predictable → predictability

You don't actually motivate why you switch from continuous rainfall to categorical precip clusters in this work. Was there a reason beyond simply proof of concept? Maybe a comment around L65 on this?

L141: Just out of curiosity, can you tell us what other regions you tested this on?

The numbering for appendix plots has gone wrong.

You mention a few times that this is a generative model: is there any value in the generative aspect here? I can't immediately think of one, but if you have thoughts perhaps share them in the conclusion?

I don't think its an important enough issue to require any changes here, but its worth bearing in mind for any future work that CCA can behave unreliably for correlated data, and that the closely related PLSR is more stable in this regard (differing only in that it maximixes cross covariance rather than cross correlation) https://arxiv.org/abs/2107.06867. Anecdotally, the scikit learn implementation of PLSR seems to handle full field data fine, so you could perhaps have avoided the district aggregation and ridge regularization process.

Clarification of the ML architecture
My main comment is that the description of the CMM-VAE architecture is quite opaque and not quite self-standing – I had to read your previous RMM-VAE paper in detail, and ultimately the RMM-VAE python code to work out what exactly was going on – and I still think I have some things wrong. I suggest some clarifications on this as follows:

Around L170 it could be useful for the readership to explicitly introduce VAEs as a generalisation of PCA as done in Murphy. E.g.

*'Where PCA deterministically maps high-dimensional input data to a low-dimensional space (which due to linearity can be interpreted as a series of patterns), VAEs map input data to a low-dimensional probability distribution, normally parameterised as a multivariate Gaussian, in a space which is not directly interpretable.'*

The big green arrow in figure 3 seems to imply that you model z|t, but you only model z|c and c|t, correct? Could you flip c and mu/sigma in the graphic to clarify that?

In appendix A can you:
a) Reiterate the interpretation of each loss term
b) Explicitly list the various predictive models that make up the CMM-VAE with a bit more exposition about the concrete details? If I have got my head round it, its something like this:

- $q(c^k|x)$= a part of the encoder neural network, predicting probability of each class. This is the only part you actually use once the model is trained.
- $q(z|x)$ = part of the encoder neural network predicts mu and sigma for each class, then (k different?) points in z are sampled from those distributions.
- $q(t|x)$= the third part of the encoder neural network, predicting precipitation class from x
- $p(x|z)$ = the decoder neural network, reproducing x from a point in z.
- $p(z|c^k)$ = another mu and sigma used to generate a point in z, but based only on the class assignment. This is a linear regression? I'm also confused why its part of p, not q, as its predicting z.
- $p(c^k|t)$= a logistic regression from precip classes onto the regimes.

A diagram (like a cleaned up equivalent of this one from your last work) would be useful:

---

## Author Response (AR1)

We thank both reviewers for the helpful and critical comments that helped us to improve the manuscript. Please find our detailed responses below.

The reviewer comments are formatted blue, our response black, and the modified text below in italics.

**Response to Review 1**

In this paper the authors address the question of how to identify useful weather regimes, which are both predictable in current forecasting systems and which do a good job at explaining variability in a target field of interest: in this case extreme wintertime rainfall over Morocco. Much ink has been spilled in the past 20 years on the topic of which weather regimes to use, how to define them, etc. I find the approach the authors have taken here refreshing and exciting. The use of a machine learning approach based on a variational autoencoder architecture to perform this task is, as far as I am aware, a novel innovation in this area, putting aside the authors' recent related paper. The focus on the practical value of the regimes for forecasting is also conceptually clarifying and provides a good motivation for the work.

The authors compare their CMM-VAE regimes to both a classical K-means approach and a surface-weather targeted CCA approach and show that their method explains the most categorical rainfall variability while remaining similarly well predicted as the classical regime, with only a slight decrease in persistence and separability. They then go on to investigate sources of predictability, linking the predictability of the regime patterns to the polar vortex and Madden Julian oscillation. I really like this work, and my only comments are editorial in nature; I believe the paper is publishable after minor revisions to increase clarity and to fix a few small typos.

Minor points:
- L55: predictable→ predictability

Corrected in the text.

- You don't actually motivate why you switch from continuous rainfall to categorical precip clusters in this work. Was there a reason beyond simply proof of concept? Maybe a comment around L65 on this?

We added a brief motivation to line 65 and following.

*The CMM-VAE method enables the application of the approach presented in Spuler et al (2024) to spatial patterns of extreme precipitation, which can provide information that is more useful at local scales compared to the spatially averaged precipitation used in Spuler et al (2024).*

- L141: Just out of curiosity, can you tell us what other regions you tested this on?

We first tested the method on the Mediterranean region used in the previous RMM-VAE paper, and on a smaller region just over the Western Mediterranean. The target variable always remained the same, spatial patterns of precipitation over Morocco.

- The numbering for appendix plots has gone wrong.

We corrected the numbering of the appendix plots.

- You mention a few times that this is a generative model: is there any value in the generative aspect here? I can't immediately think of one, but if you have thoughts perhaps share them in the conclusion?

Uses of the generative property of the CMM-VAE could include simulating 'unseen' realizations of a certain regime type from the probability distribution, for example, to explore within-regime diversity or other properties of the regimes in a larger sample size. From a given precipitation event, one could also generate possible dynamical states that led to this event, which could be another application. We don't make use of this generative property here, but have now motivated its potential use more in the sentence starting line 177.

*The method is referred to as generative because the probabilistic reduced space is continuous and can be used to simulate new realizations of the resulting regimes.*

- I don't think its an important enough issue to require any changes here, but its worth bearing in mind for any future work that CCA can behave unreliably for correlated data, and that the closely related PLSR is more stable in this regard (differing only in that it maximixes cross covariance rather than cross correlation) https://arxiv.org/abs/2107.06867. Anecdotally, the scikit learn implementation of PLSR seems to handle full field data fine, so you could perhaps have avoided the district aggregation and ridge regularization process.

In this paper, we decided not to change the comparison method, but we have added a comment in Section 2.2.2 on the fact that the regularisation of CCA was introduced in (Vinod, 1976) to handle the issue of applying CC to correlated data.

*The ridge regularization penalizes the number of dimensions used and has been shown to address numerical convergence issues of CCA when applied to collinear data.*

Clarification of the ML architecture: My main comment is that the description of the CMM-VAE architecture is quite opaque and not quite self-standing – I had to read your previous RMM-VAE paper in detail, and ultimately the RMM-VAE python code to work out what exactly was going on – and I still think I have some things wrong.

Thank you for raising this point, as well as your in-depth review and suggestions. The RMM-VAE paper was explicitly a Methods paper and thus was quite technical, so in this paper, we wanted to keep the methods section shorter and focus more on the dynamical drivers and their

predictability (which is also why it was submitted to WCD). However, you are correct that now the paper almost requires reading the RMM-VAE paper to understand the method. We therefore addressed your comment by expanding the Appendix section on the method, adding an additional figure to describe the architecture, as well as implementing your other suggestions, and believe that this enables the interested reader to understand the method fully without referring to the other paper, without distracting from the flow of the paper for the reader not interested in the details of the method.

I suggest some clarifications on this as follows:

- Around L170 it could be useful for the readership to explicitly introduce VAEs as a generalisation of PCA as done in Murphy. E.g.'Where PCA deterministically maps high-dimensional input data to a low-dimensional space (which due to linearity can be interpreted as a series of patterns), VAEs map input data to a low-dimensional probability distribution, normally parameterised as a multivariate Gaussian, in a space which is not directly interpretable.'

We added a comparison to PCA in the introductory paragraph on the CMM-VAE method, in section 2.2.3.

*A variational autoencoder (VAE) is a deep generative machine learning method used for non-linear dimensionality reduction, that is to find a reduced representation of a high-dimensional input space. Autoencoders can be interpreted as a non-linear extension of PCA implemented through an encoder and decoder neural network. Variational autoencoders extend the encoder-decoder architecture of autoencoders by fitting a probabilistic model into the reduced space using Bayesian variational inference Kingma et al. 2013.*

- The big green arrow in figure 3 seems to imply that you model $z|t$, but you only model $z|c$ and $c|t$, correct? Could you flip c and mu/sigma in the graphic to clarify that?

We edited the figure to clarify this.

In Appendix A, can you:
a) Reiterate the interpretation of each loss term
b) Explicitly list the various predictive models that make up the CMM-VAE with a bit more exposition about the concrete details? If I have got my head round it, its something like this:

- $q(c_k|x)$= a part of the encoder neural network, predicting probability of each class. This is the only part you actually use once the model is trained.
- $q(z|x)$ = part of the encoder neural network predicts mu and sigma for each class, then (k different?) points in z are sampled from those distributions.
- $q(t|x)$= the third part of the encoder neural network, predicting precipitation class from x
- $p(x|z)$ = the decoder neural network, reproducing x from a point in z.
- $p(z|c_k)$ = another mu and sigma used to generate a point in z, but based only on the class assignment. This is a linear regression? I'm also confused why its part of p, not q, as its predicting z.

- p(ck|t)= a logistic regression from precip classes onto the regimes.

A diagram (like a cleaned-up equivalent of this one from your last work) would be useful:

We added a description of the individual loss terms to Appendix A, as well as a bit more detail on the method, and the suggested figure, which has more details on the architecture.

*The loss function can be interpreted as follows: q(ck|x) is the probability of an input datapoint being part of cluster k, predicted from the encoder network. q(z|x) gives the probability distribution of latent space z given input data x, parameterized as a multivariate Gaussian with mean mu and standard deviations sigma, also predicted from the encoder network, as is q(t|x) is the prediction of the precipitation class given input x. p(z|c^k) is the latent variable as predicted from the cluster assignment ck, p(x|z) the latent space predicted from the decoder network, and p(c^k|t) the cluster assignment predicted from the target variable which is used as prior to the cluster assignment.*

*The first term of the loss function corresponds to the reconstruction loss of the dimensionality reduction, the second term to cluster coherence, i.e. penalizes the distance of points in the latent space from their assigned cluster center. The third term can be interpreted as the regularization loss, and the last term as the prediction loss of the target variable t.*

**Response to Review 2**

The paper presents a novel generative machine learning- based variational autoencoder method (CMM-VAE) designed to identify atmospheric circulation regimes associated with precipitation patterns over Morocco. The study explores the trade-off between information content of regimes of local precipitation extremes and their sub-seasonal predictability. The proposed CMM-VAE approach is compared against more traditional linear methods (PCA+k-means and CCA+k-means), demonstrating that the autoencoder-based method retains more information about precipitation while preserving the predictability offered by conventional techniques. Furthermore, the analysis examines connections with established subseasonal teleconnections, such as the MJO and stratospheric vortex variability.

Overall, this is an interesting and well-structured study that illustrates the potential of the CMM-VAE framework in identifying circulation regimes with regional impact – an important goal in regime decomposition research. The paper is clearly written, the figures are of high quality, and the results are compelling. However, I have several specific comments and questions that I recommend the authors address before the paper is suitable for publication.

Title:

The title includes the term "informative", which in the manuscript is given a technical definition based on entropy. However, the term also carries a more general, informal connotation. I suggest reconsidering the phrasing of the title to avoid ambiguity and ensure the readers do not misinterpret the intended meaning.

We agree that the interpretation of most readers would initially be the informal interpretation of the word informative. Because we believe that this informal interpretation captures the findings of the paper, we decided to keep it in the title. However, we make this distinction more explicit in the main body of the text, specifically in section 3.1. where the results on regime informativeness are presented.

Abstract:

- L3: Rossby wave packages can also act as drivers of extreme rainfall, but they are not circulation regimes per se. Consider rephrasing or clarifying.

We clarified this point by adding the word 'regional' before dynamical drivers here, but do not expand on this further in the abstract.

*In the midlatitudes, regional dynamical drivers are frequently represented as discrete, persistent and recurrent circulation regimes.*

- Which large-scale drivers are predictable but not informative?

Conventional weather regimes do not necessarily capture the dynamical processes modulating an impact variable of interest, which we discuss further in the introduction (line 40, paragraph starting line 46, and paragraph starting line 68), as well as the response to the comment further below.

*A well-established approach to representing regional dynamical drivers, such as jet variability in the midlatitudes, is the identification of recurrent and persistent patterns of atmospheric circulation, so-called weather regimes (Ghil and Robertson, 2002; Hannachi et al., 2017). These regimes are commonly identified using a combination of linear dimensionality reduction and non-probabilistic clustering (see e.g. Michelangeli et al. (1995)). […] While these conventional weather regimes are designed to capture the main features of the circulation over a given region such as the North Atlantic, they do not necessarily disentangle the dynamical patterns that modulate extreme impacts in a specific country or area of interest (Bloomfield et al., 2020; Wiel et al., 2019; Vrac and Yiou, 2010; Mastrantonas et al.,2020). […] Previous studies have shown that extreme precipitation events over Morocco are associated with dynamically driven moisture flux from the Atlantic. This can occur through an alignment of the subtropical jet with the African coastline and anomalous south-westerly surface to mid-tropospheric flow, leading to large-scale ascending motions and instability over the Western Mediterranean region (Dayan et al., 2015; Khouakhi et al., 2022; Toreti et al., 2010). These regional dynamical drivers of precipitation over Morocco have been studied in terms of both North Atlantic and Mediterranean circulation regimes (Driouech et al., 2010; Mastrantonas et al., 2020; Tramblay et al., 2012). While certain regimes over both regions, such as the negative phase of the North Atlantic Oscillation (NAO), are associated with an increase in the probability of extreme precipitation, the dynamical mechanisms described above and analysed in terms of distinct patterns in Chaqdid et al. (2023) are not clearly captured in the regimes over either region.*

- Is the physical interpretability an inherent outcome of the method, or is it more a feature of this specific application? Could other methods also yield interpretable regimes?

The physical interpretability is an inherent outcome of the method and is mentioned here in particular to contrast it with methods such as analysis of composites during precipitation extremes shown below. Even though the pattern below is (by construction) informative of extreme precipitation over Morocco, as it is the composite of days with extreme precipitation, there are different dynamical configurations of the jet that could lead to this pattern, as shown in the targeted regimes in Figure 4 - it is therefore not dynamically interpretable.

[Figure]

*Figure 1: ERA5 1981-2022 z500 composites during extreme precipitation days (total precipitation exceeds 90th percentile) - pre-processing as described in the manuscript.*

- L46: The claim that regional regimes do not modulate local extremes is not fully convincing. Two studies alone may not be sufficient to generalise this conclusion. Is it plausible that appropriate downscaling techniques (e.g., Baker et al., 2018) could reveal such relationships? Please elaborate or qualify the statement.

We are stating the following claim: "While conventional weather regimes are designed to capture the main features of the circulation over a given region such as the North Atlantic, they do not necessarily disentangle the dynamical patterns that modulate extreme impacts in a specific country or area of interest (Bloomfield et al., 2020; Wiel et al., 2019)." We believe that this claim is supported by a range of different papers studying surface-level anomalies associated with conventional regimes - see below for example the following figure from *van der Wiel et al. (2019): The influence of weather regimes on European renewable energy production and demand,* where it can be seen that the NAE regimes (left plot below) are informative only over parts of Europe for the variable 10m wind and 2m temperature (composites in the right plot below).

[Figure]

By "conventional weather regimes", as qualified in the sentence prior to the one cited above, we mean regimes that are identified based on dimensionality reduction and/or clustering, without making use of the target variable ("These regimes are commonly identified using a combination of linear dimensionality reduction and non-probabilistic clustering (see e.g. Michelangeli et al. (1995)). "). We have added two additional papers that explicitly mention this point (citations see below).

Baker et al. (2018) use a "linear combination of two mean sea-level pressure (MSLP)-based indices which are derived from the MSLP correlation patterns for precipitation in northwest Scotland and southeast England" to construct a regression model from circulation to precipitation - so they do incorporate information about the target variable to construct a suitable MSLP index. Thus, their study is a case in point that making use of the target variable is useful, and we now cite the paper in line 35.

References:

Baker, L., L. Shaffrey & A. Scaife (2018). Improved seasonal prediction of UK regional precipitation using atmospheric circulation. Int. J. Climatology, 38, e347-e453.

Vrac, M. and Yiou, P.: Weather regimes designed for local precipitation modeling: Application to the Mediterranean basin, Journal of Geophysical Research, 115, D12 103, https://doi.org/10.1029/2009JD012871, 2010.

Mastrantonas, N., Herrera-Lormendez, P., Magnusson, L., Pappenberger, F., and Matschullat, J.: Extreme precipitation events in the Mediterranean: Spatiotemporal characteristics and connection to large-scale atmospheric flow patterns, International Journal of Climatology, 41, 2710–2728, https://doi.org/10.1002/joc.6985, https://onlinelibrary.wiley.com/doi/pdf/10.1002/joc.6985, 2020

- L75: Specify which mechanisms in terms of distinct patterns are not captured in the regimes?

In the introduction, we specify the dynamical driver modulating extreme precipitation over Morocco that is not captured by the conventional NAE regimes (see below). The findings of the paper then show specific distinct patterns that modulate precipitation over Morocco but are not captured in the North Atlantic regimes, but these are shown in the results section 3.1. and discussed in the final section 4.

*Previous studies have shown that extreme precipitation events over Morocco are associated with dynamically driven moisture flux from the Atlantic. This can occur through an alignment of the subtropical jet with the African coastline and anomalous south-westerly surface to mid-tropospheric flow, leading to large-scale ascending motions and instability over the Western Mediterranean region (Dayan et al., 2015; Khouakhi et al., 2022; Toreti et al., 2010). These regional dynamical drivers of precipitation over Morocco have been studied in terms of both North Atlantic and Mediterranean circulation regimes (Driouech et al., 2010; Mastrantonas et al., 2020; Tramblay et al., 2012).*

- L90: Clarify the comparative advantage over Spuler et al (2024a). How does using categorical target variables enable the use of CHIRPS rather than reanalysis precipitation? Also, explain what "categorical variables" mean in this context.

Thank you for pointing this out; the structure made the sentence misleading. We rewrote the sentence(s) to clarify this point. We thereby also eliminated the term 'categorical variables', which we now introduce only in the Methods section, instead referring directly to the spatial precipitation patterns here.

*The contribution of this paper is threefold. The first contribution is to introduce the CMM-VAE method, short for Categorical Mixture Model Variational Autoencoder, which enables the application of the method previously presented in Spuler et al. (2024a) to study spatial patterns of precipitation extremes as target variables. Furthermore, we work with a more realistic*

*precipitation dataset (CHIRPS) instead of the reanalysis-based precipitation data used in Spuler et al. (2024a).*

Data and Methods:

- Fig 2: Please explain the vertical axis in panel b more clearly. What does it represent? Could Fig 2 also indicate the subregions or district-level divisions referenced in Table 1, especially for the CCA+k-means method?

We added a more detailed explanation of the vertical axis in Figure B to the figure caption, see below.

Revised figure caption: *Top row: spatial patterns of 3-day precipitation events over Morocco in extended winter (November - March) identified using k-means clustering. The percentage number in the heading indicates the occurrence probability over all days. Bottom row: occurrence probability of clusters in different quantiles of total precipitation over Morocco. The vertical axis represents the percentage of days in a given precipitation quantile that are assigned to a specific pattern.*

As we didn't want to overcrowd the figure or add an additional figure in an already full appendix, we reference Loudyi et al 2022, Figure 2.1, who used the same administrative boundaries.

- L154: When citing "the eofs package (Dawson, 2016)", a brief description would be helpful.

We added the specification that this is a Python package we are citing, and believe the method implemented in the package is described in the paragraph above.

- L164: Explain the role of the ridge regularization parameter

We added an additional sentence explaining the role of the regularization parameter (line 166).

*The ridge regularization penalizes the number of dimensions used and has been shown to address numerical convergence issues of CCA when applied to collinear data.*

Predictability metrics:

- L213: The stated start dates of 22/03 likely refers to 22/02, as forecasts for April fall outside the defined extended winter period Nov- March. Please correct if this is a typo.

We did intend to write 22/03. Because we are evaluating lead times 0-46, we use the first 10 days predicted from the 22/03 initialization.

- L225: An ensemble of 11 members is relatively small, potentially affecting probability estimates and skill scores. Have you considered using fair scores to mitigate bias? How

might ensemble size impact the results? Which of the findings are robust in light of this limitation?

We considered using fair scores. Because all three methods, PCA, CCA and CMM-VAE are evaluated on the same number of ensemble members, we came to the conclusion that this would not make a difference for the comparison of the methods, which is the focus of this paper. Given that the same number of ensemble members is available in either method, we believe that the block-bootstrapping approach is adequate for investigating the robustness of our findings.

Results:

- L335: Given the confidence intervals, the difference in BSS drop-off between PCA and CMM-VAE (17 vs 19 days) is likely not statistically significant. The text should acknowledge that their performance of BSS and ROC AUC is generally similar, with only CCA clearly underperforming.

We took out the part of the sentence specifying 17 vs 19 days as this would, as you say, have to come with a statement on the significance of this difference.

The sentence now reads: *The targeted CMM-VAE regimes are found to be as predictable in terms of both BSS and ROC-AUC as the non-targeted regimes identified using PCA + k-means.*

- L340: Clarify what is meant by "slightly" – does this imply that differences are statistically insignificant?

By "slightly" we mean that both reliability and resolution of both methods are within the confidence interval of the respective other, but there is a small difference in the mean. We clarify this in the revised text.

*We find that the CMM-VAE regimes perform slightly worse in terms of resolution but slightly better than PCA + k-means regimes in terms of reliability, i.e. the reliability and resolution of both methods are within the confidence interval of the respective other and there is a small difference in the mean.*

- L361: Sentence structure appears incomplete; a verb may be missing.

Corrected in the text.

- L389: While conditional entropy is clearly lowest for CMM-VAE at longer leads, the mutual information metrics shows less pronounced differences, given the uncertainty. Please mention this and consider discussing potential reasons for the divergence between mutual information and entropy metrics for PCA vs CMM-VAE at long lead times.

As this difference can also be seen in teleconnections from the Stratospheric Polar Vortex discussed in 3.3.1, we provide possible reasons for this there:

*We find that conditional entropy, which quantifies the average uncertainty in the targeted circulation regime remaining given the state of the polar vortex is known, is similar for the PCA + k-means and CMM-VAE regimes and higher for the CCA regimes. Mutual information between the regimes and SPV states, which also takes into account the uncertainty in the regimes themselves, is highest for the PCA + k-means regimes and lowest for the CCA regimes. The difference between conditional entropy and mutual information is due to the fact that the entropy of the PCA + k-means regimes appears to be larger than that of the CMM-VAE regimes, which can vary depending on the lead time analysed in the hindcasts.*

Discussion and Conclusions:

- Why does the targeted CCA+k-means method underperform relative to the non-targeted PCA+k-means? Please provide a hypothesis or possible explanation.

In the Results section, we mention a possible reason for this in line 307. We have now added a comment on this in line 420 in the Discussion section.

*The CCA + k-means method projects the geopotential height data into a subspace which is maximally correlated with precipitation over Morocco, thereby identifying targeted clusters which are associated with an increase in extreme precipitation over Morocco but which show less structure in the rest of the atmospheric phase space.*

*The lower predictability of the targeted CCA + k-means regimes can be attributed to the fact that the method projects the data into a correlated subspace but does not capture the structure in the rest of the phase space as well.*

- Could your conclusions be sensitive to the chosen target domain size? A brief discussion on this would be valuable.

We believe that as the conclusions are currently written, we do not overstate the findings of this paper and explicitly leave the generalizability of these findings to future work ("The results imply that *in this region*, the CMM-VAE method is able to identify a representation of regional dynamical drivers that balances and even resolves the trade-off between informativeness of local impacts and subseasonal predictability." [...] "The method could be further tested through application to further regions and target variables."). Ongoing research investigating the application of this method to study heat extremes over Northern Europe does confirm conclusions drawn in this paper for this other target region and variable, but we leave it to this other paper to draw that conclusion.

- L437: While you note that the study disentangles the role of the two drivers influencing Moroccan precipitation, can you also infer any potential amplifying effects when both

drivers are active? Even if a detailed quantification is beyond the scope here, some speculative discussion would enhance the conclusions.

Because an individual circulation state will be assigned to one of the two drivers or an intermediate state, the two regimes themselves would not have amplifying effects. However, there are many other effects that would act to amplify the large-scale circulation-driven precipitation. We added a comment on these amplifying effects in the discussion, as the primary focus of this paper is on the large-scale circulation.

*The method could be further tested and refined through application to further regions and target variables, and potential amplifying effects with local drivers of precipitation could be investigated.*

---

## Author Response (AR2)

Changes made to the document 31/07/25:

- Corrected the placement of sections.
- Added link to access the code to reproduce the results in the section Data and Code Availability.
- Corrected label in Figure 7 (no changes to the Figure itself).
- Reorganized Acknowledgements and Author Contribution sections.